# Interaction between miR4749 and Human Serum Albumin as Revealed by Fluorescence, FRET, Atomic Force Spectroscopy and Computational Modelling

**DOI:** 10.3390/ijms23031291

**Published:** 2022-01-24

**Authors:** Valentina Botti, Silvia Marrone, Salvatore Cannistraro, Anna Rita Bizzarri

**Affiliations:** Biophysics and Nanoscience Centre, Department of Ecology and Biology DEB, Università della Tuscia, Largo dell’Università, 01100 Viterbo, Italy; valentina.botti@unitus.it (V.B.); s.marrone.95@gmail.com (S.M.); cannistr@unitus.it (S.C.)

**Keywords:** miR4749, Human Serum Albumin, fluorescence quenching, FRET, computational docking

## Abstract

The interaction of Human Serum Albumin (HSA) with the microRNA, miR4749, was investigated by Atomic Force Spectrscopy (AFS), static and time-resolved fluorescence spectroscopy and by computational methods. The formation of a HSA/miR4749 complex with an affinity of about 10^4^ M^−1^ has been assessed through a Stern–Volmer analysis of steady-state fluorescence quenching of the lone Trp residue (Trp214) emission of HSA. Förster Resonance Energy Transfer (FRET) measurements of fluorescence lifetime of the HSA/miR4749 complex were carried out in the absence and in the presence of an acceptor chromophore linked to miR4749. This allowed us to determine a distance of 4.3 ± 0.5 nm between the lone Trp of HSA and the dye bound to miR4749 5p-end. Such a distance was exploited for a screening of the possible binding sites between HSA and miR4749, as predicted by computational docking. Such an approach, further refined by binding free energy calculations, led us to the identification of a consistent model for the structure of the HSA/miR4749 complex in which a positively charged HSA pocket accommodates the negatively charged miRNA molecule. These results designate native HSA as a suitable miRNA carrier under physiological conditions for delivering to appropriate targets.

## 1. Introduction

Human serum albumin (HSA) is the most abundant globular protein in the plasma, accounting for 75% of the colloid osmotic pressure of blood [1,2,3,4,5]. HSA is also the prevalent protein in interstitial fluids, with various concentrations in different body compartments [4,6]. HSA exerts a variety of physiological functions, among which are provision of most extracellular antioxidant activity, regulation of intracellular and plasma pH [7] and inhibition of proinflammatory pathways involving tumor necrosis factor α and nuclear factor κB [6]. However, the most exploited feature of HSA resides in its versatility as a transport protein, since it is able to non-covalently bind and deliver to its targets an extraordinarily diverse range of ligands, varying from long chain fatty acids, hormones, nutrients [8] and metal ions [9], to bile acids, nitric acid and endotoxin [6], inter alia. Due to its natural properties as a ubiquitous and versatile carrier, combined with its biocompatibility and easy nano-engineering, HSA and HSA-based nanovectors were extensively explored for delivery of systemic diagnostic and therapeutic compounds [3]. HSA binding significantly influences pharmacokinetics by increasing solubilization of hydrophobic drugs, as well as increasing their half-life in biologic fluids [10,11]; it can also result in a reduction in their pharmaceutical effectiveness [12]. Moreover, it has been reported that carbamylated HSA can upregulate the expression of a microRNA and, in particular, miR-146a/b in human renal cell carcinoma [13]. MicroRNAs (miRNAs) are short (average of 22 nucleotides) non-coding single stranded RNAs, which can post-transcriptionally regulate the expression of genes controlling fundamental cell life processes, such as cell proliferation, differentiation, stress response and apoptosis. Indeed, they induce translational repression or activation through the binding of target messenger RNAs [14,15,16,17,18,19,20].

At present, over 2600 mature miRNAs have been identified [21], with their gene down/up regulation mechanisms still being not fully understood. Different miRNAs have specific stable levels of expression depending on tissues and on developmental stage and, when dysregulated, they can contribute to various pathophysiologies, as observed in leukemia, cardiovascular and nervous system disorders, inflammatory diseases and, especially, cancers [22,23,24]. Indeed, miRNAs can influence tumorigenesis by modulating the activity of both oncogenes and tumor suppressor genes, and they can promote tumor progression and metastasis since they are actively secreted in the extracellular spaces to be transferred between cells [25,26,27,28]. For these reasons, miRNAs have emerged as promising diagnostic and prognostic biomarkers [29] as well as therapeutic targets and agents. Consistent with their role in intercellular communication, miRNAs have been found circulating in various HSA-rich body fluids, including serum and interstitial fluids, carried by extracellular vesicles and exosomes and bound to lipids or proteins of the AGO family [30]. Some studies involving HSA and miRNAs have focused on modification of HSA (covalent binding/complexation with cationic polymers, thiolation), or coating and functionalization of HSA nanoparticles, in order for it to serve as a gene delivery platform ([31] and articles cited therein) or as a scaffold that hosts receptors and probes for miRNAs’ detection [32,33]. However, basic information on miRNAs’ interaction with HSA is missing, despite it being considered an essential parameter [12] on which a species’ activity and detectability crucially depend.

Hence, here we have investigated the possibility that native, unmodified HSA could bind miRNAs largely present in the cell environment. Preliminarily, we have selected miR4749-5p (miR4749) as a relevant representative candidate. miR4749 has been found to exhibit altered expression in hepatocellular carcinoma [34], binding to the long non-coding RNA MAPKAPK5-AS1, which plays a critical role in carcinogenesis, although its function is still unknown. It is also abnormally expressed in rectal cancer [35] and polycystic ovary syndrome [36]. Moreover, miR4749 has been observed to have a tumor suppressive role in mediating glioblastoma progression by interacting with Replication Factor C subunit 2 [37]. Recently, our group found that miR4749 forms a complex with the DNA binding portion (DBD) of the tumour suppressor p53, with this result indicating a correspondence between the high sequence similarity of miR4749 with the DNA Response Element (RE) of p53 family [38].

In the present study, the lone intrinsic Tryptophan (Trp214) of HSA has served both as a solvatochromic fluorophore in fluorescence quenching measurements and as an energy donor in FRET experiments which, assisted by molecular modelling and free energy calculations, allowed it to establish the formation of a specific HSA/miR4749 complex and to provide information about the structure of this complex. To further check the formation of a complex, we have applied Atomic Force Spectroscopy (AFS), a technique, operating under near physiological conditions, able to provide information about the intermolecular forces and the kinetic properties of a biomolecular complex [39,40,41]. Our results confirm that HSA can bind and transport a miRNA molecule under physiological conditions, suggesting that its role in the circulation and bioavailability of miRNAs cannot be overlooked and needs further investigation to be fully understood and, possibly, exploited in therapeutics and diagnostics.

## 2. Results

### 2.1. Fluorescence Results

The interaction between HSA and miR4749 was first investigated by fluorescence spectroscopy. Steady-state emission spectra of 5µM HSA solutions in PBS buffer, alone and with progressively higher concentrations of miR4749 (up to a 3:1 ratio), are shown in Figure 1A. Spectra were acquired by using an excitation wavelength (λ_exc_) of 295 nm, which allowed selective monitoring of the fluorescence profile of the lone tryptophan residue of HSA, Trp214, since all the other fluorophores (Tyr and Phe aromatic residues) have negligible absorbance at this wavelength. Moreover, no interfering fluorescence signal from miR4749 was detectable in the relevant emission spectral range (310–500 nm).

HSA showed maximum emission intensity at 346 nm (Figure 1A, black line), compatible with Trp214 being almost fully exposed to the solvent [42]. Increasing additions of miR4749 (coloured lines in Figure 1A) induced a progressive reduction in HSA’s fluorescence. No wavelength shift of the emission peak was observed, signalling that the presence of miR4749 did not affect the folding of HSA around the Trp214 residue. In order to rationalize the observed quenching behaviour and to extract interaction parameters, fluorescence quenching data were first analysed through a Stern–Volmer Equation:F_0_/F = 1 + k_q_ τ_q_ [Q] = 1 + K_SV_ [Q](1)
where F_0_ and F are the steady-state fluorescence intensities at 346 nm of HSA in the absence (F_0_) and in the presence (F) of quencher Q (here miR4749), k_q_ is the bimolecular quenching constant, τ_q_ is the lifetime of Trp214 in the absence of a quencher and K_SV_ is the Stern–Volmer quenching constant. The obtained plot of F_0_/F vs. [Q], shown in Figure 1B, substantially follows a linear trend. A fit by Equation (3) allowed us to extract the Stern–Volmer constant, K_SV_ = (2.7 ± 0.3) × 10^4^ M^−1^. The changes in fluorescence intensities upon adding the quencher were also investigated by a nonlinear Stern–Volmer model, which takes into account the presence of other effects than quenching [43]. In particular, the quantity (F_0_ − F)/F_0_ was analysed by the following equation [44,45]:(2)$$\frac{F_{0}-F}{F_{0}}=\frac{1}{2{\left[HSA\right]}_{0}}[\left[\frac{1}{K_{SV}}+{\left[HSA\right]}_{0}+{\left[Q\right]}_{0}\right]]\;-\sqrt{{\left(\frac{1}{K_{SV}}+{\left[HSA\right]}_{0}+{\left[Q\right]}_{0}\right)}^{2}-4{\left[HSA\right]}_{0}{\left[Q\right]}_{0}}$$
where [HSA]_0_ is the initial concentration of the fluorophore while [Q]_0_ is the initial concentration of the quencher, which is assumed to be equal to the free quencher concentration. The plot of (F_0_ − F)/F_0_ (blue squares), shown in the inset of Figure 1B, is characterized by a slight deviation from a linear trend; a good description being obtained by Equation (4) (see the blue line). The extracted K_SV_ of (3.0 ± 0.3) × 10^4^ M^−1^ is in a good agreement with the value determined by the linear Stern–Volmer approach. This indicates that, at our conditions, the linear Stern–Volmer model gives a thorough description of the process, and that K_SV_ provides a reliable estimation of the affinity.

To gather further indications about the interaction, measurements by time-resolved fluorescence spectroscopy were carried out. The decays of HSA alone and HSA in presence of miR4749 at a 1:1 ratio were analysed by Equations (1) and (2) (see Section 3). The average fluorescence lifetime evaluated for HSA alone, τ_q_, is of (5.43 ± 0.03) ns, in good agreement with the literature [46]. Relevantly, we observed that τ_q_ was maintained after additions of increasing concentrations of miR4749, with deviations less than 2% and a τ = (5.33 ± 0.06) ns registered for the 1:1 ratio, which constitutes evidence for the occurrence of static quenching. Furthermore, the bimolecular quenching constant, k_q_, derived from K_SV_/τ_q_, amounts to (4.8 ± 0.6) × 10^12^ M^−1^ s^−1^, two orders of magnitude higher than the typical diffusion controlled quenching rate, with this also pointing to a static quenching mechanism [43]. Such combined findings support the formation of a specific complex between HSA and miR4749 in the ground state [43], with an affinity constant, K_A_, accounted by K_SV_.

Concerning the interaction mode, an allosteric mechanism is suggested by the observed maintenance of the emission peak wavelength: the binding of the microRNA yields a conformational change of HSA without altering Trp214′s exposition to the solvent.

### 2.2. AFS Results

The interaction between HSA and miR4749 was also investigated by applying AFS, using a miR4749-functionalized tip and a HSA-conjugated substrate, as described in Section 3.3. Approach–retraction force curves were collected at five increasing loading rates, r. For each loading rate, the unbinding forces, corresponding to specific unbinding events, were evaluated and cast into a histogram. In all the cases, we found a single mode distribution whose maximum provides an estimation of the most probable unbinding force (F*); a representative histogram is shown in Figure 2A.

We note that the molecular dissociation, detected in AFS measurements, occurs under the application of an external force and then far from the thermodynamic equilibrium. On such a basis, suitable theoretical models have been developed to extract the kinetic and energy landscape parameters at the equilibrium [47]. We used the Bell and Evans model [48,49], which predicts a linear dependence of the F* on the natural logarithm of the loading rate, through the following expression:(3)$$F^{*}=\frac{k_{B}\;T}{x_{\beta }}\ln\frac{r\;x_{\beta }}{k_{off}\;k_{B}\;T}$$
where k_B_ is the Boltzmann constant, T is the absolute temperature, k_off_ is the dissociation rate constant and x_β_ is the width of the energy barrier along the direction of the applied force. The plot of F* vs. the logarithm of the loading rate exhibits a single linear trend indicative of a single energy barrier for the reaction (see Figure 2B). A fitting of unbinding forces by Equation (5) led us to determine k_off_ = (3.0 ± 0.6)⋅10^−1^ s^−1^ and x_β_ = (0.26 ± 0.08) nm. Both the x_β_ and k_off_ values are rather close to those found for the interaction of the DNA binding portion of p53 with a different miRNA [50]. To extract the affinity of the HAS/miR4749 complex, we estimated the association rate constant (k_on_) by following the procedure given in [51]. We found a k_on_ of ~10^4^ M^−1^ s^−1^, a value close to that derived for other biomolecular systems [50,51]. Accordingly, an affinity constant K_A_= k_on_/k_off_, of ~3.0 10^4^ M for the HSA/miR4749 complex, was determined, this value being in agreement with that estimated by fluorescence.

### 2.3. FRET Results

In order to explore the structure of the HSA/miR4749 complex, FRET was employed as a guide to the localization of the binding site. Namely, the distance R between D and A can be determined from E_FRET_, i.e., the long-range non-radiative energy transfer from Trp214 (the Donor, D) to an Acceptor (A), the Atto390 dye, attached to the 5′ end of miR4749, through the relation [43]:E_FRET_ = R_0_^6^ / (R_0_^6^ + R^6^)(4)
where R_0_ is the Förster radius, i.e., the D–A distance at which E_FRET_ assumes the value of 0.5 [52].

The E_FRET_ was evaluated by the donor lifetime variation method [53], according to [43]:E_FRET_ = 1 − (<τ_DA_> / <τ_D_>)(5)
where <τ_D_> is the average fluorescence lifetime of Trp214 when miR4749 is bound to HSA (1:1 ratio), while <τ_DA_> is its lifetime when the binder is bearing the acceptor (HSA/miR4749–Atto390 (1:1) complex). Atto390 was chosen because its absorbance overlaps the emission spectrum of Trp214 [50], a condition required for the dipole–dipole coupling enabling FRET.

As reported in Figure 3, <τ_D_> was found to be (5.33 ± 0.06) ns, while a significantly shorter fluorescence lifetime was detected in the presence of the acceptor, <τ_DA_> = 5.12 ± 0.02 ns, consistent with the occurrence of FRET. Applying Equation (7), an average value of (4.3 ± 0.5) nm was derived for the R distance in the HSA/miR4749 complex.

### 2.4. Computational Docking

Fluorescence and FRET results indicate the formation of a static complex between HSA and miR4749, with a distance of (4.3 ± 0.5) nm between Trp214 of HSA and the dye (Atto390) bound to the 5′ end of miR4749. The knowledge of this distance provides a valuable structural parameter, helping to restrict possible interaction regions between the partners and to eventually propose the topological structure of the complex.

By following the procedure described in Section 3.3, a computational docking between the X-ray structure of HSA and the best model for miR4749, shown in Figure 8A,B, respectively, was applied. Fifty tentative models for the HSA/miR4749 complex were extracted. A preliminary screening of all these models was performed by measuring the distance, D_DA_, between the center of the aromatic rings of the lateral chain of Trp214 of HSA and the 5′ end of miR4749. We found that the D_DA_ distance spans from 1 nm to 6 nm, with the largest part of complexes (more than 70%) being characterized by a D_DA_ value between 3 and 5 nm. By taking into account the calculated D_DA_ distance in the 3.8–4.8 nm interval and an estimated contribution of 0.1 nm, as due to the dye attached to the 5′ end of miR4749, we selected those complexes whose D_DA_ distance falls in the 3.9–4.9 nm interval. The resulting models were then grouped by evaluating their structural differences in terms of the RMSD. Models differing among them for RMSD values less than 0.1 nm were grouped together; the first ranked model was taken as a representative of the corresponding group. Such a screening procedure allowed us to finally select six models, named Models 1–6, collectively represented in Figure 4.

In Model 1 and Model 5, miR4749 binds at almost the same region, located between IB and IIIA subdomains, while in Model 2 it binds close to this region. Additionally, in Model 3 and Model 6, miR4749 binds close to IB and IIA. Finally, in Model 4, miR4749 binds between the IIB and IA subdomains.

To assess the stability of the formed complexes, as well to evaluate the corresponding binding free energy, all the models were submitted to a MD simulation procedure, constituting a preliminary equilibration of 3 ns followed by a 10 ns long run for data collection (with three replicates for each model (see also Section 3.4)). The temporal evolution from representative MD simulated runs of the D_DA_ distance during 10 ns is shown in Figure 5. In all the cases, the distances exhibit some oscillations superimposed on fast fluctuations; however, they remain rather close to the initial value in the analyzed time interval. The most significant deviations are detected for Model 3 and Model 4 in which the distance slightly decreases, although remaining within the selected distance range. Similar trends were obtained for the other runs. The initial and the average structures over the MD run are reported in Table 1. These data are substantially consistent with a good stability of the formed complexes.

Successively, the binding free energy, ΔG_B_, as described in Section 3.5, was evaluated for each model; the final ΔG_B_ value together with the contribution from the various analyzed components are reported in Table 1.

The nonpolar solvation term ΔG_nonpol solv_ is rather small and always negative for all the complexes. At the same time, the entropic term (−TΔS_MM_) value is rather similar for all the models and always positive. The internal energy term, ΔE_MM_, is negative, with significant differences, however, among the six complexes. Finally, a high contribution to the binding free energy comes from the electrostatic term (ΔG_pol solv_) which, however, assumes both negative and positive values in the six models this term being crucial for determining the overall binding free energy. Only for Model 1 and Model 4, is this term ΔG_pol solv_ negative, leading to a final negative ΔG_B_ and then indicating energetically favorable bound states.

To closely address the role of the electrostatic forces in the formation of the complex, the electrostatic surface potential was evaluated by APBS [54]. The electrostatic surface potentials of isolated miR4749 and HSA are shown in Figure 6A. miR4749 is characterized by an almost uniform red surface, indicative of a global negative charge. On the other hand, HSA exhibits both positive and negative charges at its surface. Interestingly, the binding site of miR4749 on HSA, indicated by an arrow, is located at a positive HSA pocket (blue colour). The final electrostatic surface potential of the HSA/miR4749 complex can be clearly visualized in Figure 6B; the position of miR4749 within the complex is highlighted in Figure 6C, in which its skeleton is represented as a green cartoon.

Similarly, even in Model 4, the binding of miR4749 occurs at the top of HSA with the involvement of a small pocket, again characterized by positive charges exposed to the solvent. Such a result strongly supports an electrostatic guide for the formation of the complex between HSA and miR4749. On the basis of these results, we selected Model 1 exhibiting the lowest ΔG_B_ value as the best model for the HSA/miR4749 complex. A front and back graphical view of this model is shown in Figure 7, with the distance between the aromatic ring center of the lateral chain of Trp214 at the 5′ end of miR4749 being indicated (see the red line and the corresponding label). The binding of miR4749 to HSA largely involves the IB domain, touching also on the IA and IIIB domains.

In summary, these results clearly indicate that HSA can form a stable complex with miR4749, with a significant contribution from the electrostatic interactions. Furthermore, it could be hypothesized that the localized positively charged regions in HSA could also bind other negatively charged miRNAs.

## 3. Materials and Methods

### 3.1. Materials

Human Serum Albumin (HSA) (molecular weight 66.5 kDa) was purchased from Sigma–Aldrich Co. (St. Louis, MO, USA) as a globulin free (purity degree >99%) lyophilized powder and used without further purification.

Human miR-4749-5p (sequence: UGCGGGGACAGGCCAGGGCAUC; 7.15 kDa; hereafter miR4749), alone and labeled at the 5′ end with the fluorescent dye Atto-390 (7.65 kDa; hereafter miR4749Atto390), were purchased from Metabion International AG (Planegg, Germany) and stocked at 253 K. The purity of miR4749 was verified by HPLC by mass spectroscopy by the producer. The medium, a 50-mM phosphate-buffered saline (PBS) solution at pH 7.4, was prepared using reagents from Sigma–Aldrich Co.

### 3.2. Spectroscopic Methods

Steady-state fluorescence measurements were conducted at room temperature using a FluoroMax^®^-4 Spectrofluorometer (Horiba Scientific, Jobin Yvon, Palaiseau, France) operated by FluorEssence software (Horiba Scientific, Jobin Yvon, Palaiseau, France). Optical-path quartz cuvettes of 1 cm were used. Emission profiles were collected in the 305–580 nm range, with increments of 1 nm and an integration time of 0.50 s under excitation at 295 nm, and setting a 5-nm band-pass width for both excitation and emission paths. Spectra were acquired in signal to reference (S/R) mode in order to minimize random fluctuations in the intensity of the Xenon lamp and corrected for the Raman signal of the buffer. Correction for inner field effect was performed. For each sample, measurements were registered in quintuplicate at regular delays (120 s) for statistical significance.

Fluorescence lifetime measurements were carried out at 298 K, equipping the FluoroMax^®^-4 Spectrofluorometer with a pulsed nanoLED source (Horiba Scientific, JobinYvon) for implementation of the TC single-photon counting method. The apparatus operated in reverse mode at a repetition rate of 1 MHz, shining an excitation light at 295 nm with a temporal width lower than 1 ns and a band-pass width of 5 nm. The samples of the investigated system were located in 1-cm optical-path quartz cuvettes. Fluorescence data were detected at 346 nm and processed by DAS6 software (Horiba Scientific, Jobin Yvon, Palaiseau, France) as a convolution of the registered impulse response function (scattered light) and the intensity fluorescence decays through the expression:(6)$$I\left(t\right)=a_{0}+\sum_{i=1}^{n}a_{i}e^{\left(-t/\tau_{i}\right)}$$
in which *I*(*t*) is the time-dependent intensity, *a*_0_ gives the background and *a*_i_ are pre-exponential factors representing fractional contribution to the time resolved decay of the *i*th component with lifetime *τ_i_*. The goodness of the fit was judged in terms of both χ^2^ value and weighted residuals. Average fluorescence lifetime, *<τ>*_i_, (out of five replicas for each sample) was then calculated by:(7)$$<\tau >=\frac{\sum_{i=1}^{n}a_{i}\tau_{i}}{\sum_{i=1}^{n}a_{i}}$$
considering two exponential contributions.

### 3.3. AFS Experiments

Functionalization of tips and substrates used in AFS experiments was performed by following the procedures reported in [50,55]. Briefly, silicon nitride AFM tips (cantilever B, MSNL-10; Bruker Corporation, Billerica, MA, USA), with a nominal spring constant, k_nom_, of 0.02 N/m, were first functionalized with silane and then with N-hydroxysuccinimide–polyethyleneglycol–maleimide (NHS-PEG-MAL, 3.4 kDa, N = 24; hereafter PEG) (Thermo Fisher Scientific, Waltham, MA, USA). The amino-reactive group (NHS) of the PEG was then coupled with the NH_2_ ends of the NH_2_-miR4749 (1.3 µM). Aldehyde-functionalized glass surfaces (PolyAn GmbH, Berlin, Germany) were incubated with 50 µL of HSA (2 µM) in PBS buffer overnight at 4 °C to promote a covalent binding of proteins via their exposed amino groups. Unreacted groups of both tips and substrates were passivated by incubation with 1 M ethanolamine hydrochloride at pH 8.5 (GE Healthcare). All the samples were stored in PBS buffer at 4 °C.

AFS measurements were performed at room temperature with the Nanoscope IIIa/Multimode AFM (Veeco Instruments, Plainview, NY, USA) in PBS buffer. Force curves were acquired by approaching tips and substrate functionalized with miR4749 towards the HSA-functionalized substrate by applying a ramp size of 150 nm. Once the preset maximum contact force value between the tip on the protein-functionalized substrate (here set at 0.7 nN) was reached, the cantilever was stopped and retracted from the substrate. During the approaching phase, if two partners have enough flexibility and re-orientational freedom to assume a correct reciprocal orientation, they undergo a biorecognition process, forming a specific complex. As retraction continues, the spring force overcomes the interacting force and the cantilever jumps off, leading to an unbinding of the complex. The exerted force, derived by multiplying the cantilever deflection at the jump-off by its effective spring constant (k_eff_), is called unbinding force, F, of the complex.

Force curves were collected by approaching the tip to different points of the substrate at a constant velocity of 50 nm/s, while the retraction velocity was varied from 50 to 4200 nm/s. The effective loading rates were calculated from the product between the pulling velocity, v, and the spring constant of the entire system, k_syst_ [56]. At each loading rate, thousands of force curves were acquired to guarantee information with statistical significance. Curves characterized by stretching features of the PEG linker during the retraction phase were selected as specific unbinding events [57]. Ambiguous unbinding events were also analyzed by using the 1/f noise approach [58]. To check the specificity of the interaction, a blocking experiment was carried out by repeating the AFS experiments using the same miR4749 functionalized tip against an HSA-functionalized substrate which was previously incubated with miR4749. The ratio of the number of events corresponding to specific unbinding events over the total recorded event is reduced from about 28% to 11% upon blocking, thus witnessing the specificity of the HSA–miR4749 interaction.

### 3.4. Modelling Procedures

Initial atomic coordinates of HSA were taken from the X-ray structure at 2.5 Å resolution (chain A of 1AO6 entry from the protein data bank) [59]. HSA is composed by a single a-helix chain of 582 amino acids organized in three repeated homolog domains (sites I, II and III), with each domain comprising two separate sub-domains (A and B). A graphical representation of HSA is shown in Figure 8A.

The initial coordinates of miR4749, not available, were obtained by the modelling procedure developed in ref. [38]. Briefly, from the sequence (reported above), the secondary structure of miR4749 as well the corresponding dot-bracket notation were derived by RNAFOLD under default parameters [60]. These data were then submitted to the SIMRNA software [61]. The selected best model for 3D miR4749 structure is shown in Figure 8B. Successively, a computational docking between this model for miR4749 and HSA was carried out by HDOCK [62]; for each ligand–receptor couple, the first ten ranked models were taken into consideration for further analysis (see below). All the structure figures were created by Pymol [63] and VMD [64].

### 3.5. Molecular Dynamics (MD) Simulations

MD simulations of HSA, miR4749 and the HSA/miR4749 complex in water were carried out by the GROMACS 2018 package [65], using AMBER03 Force Field for the protein and miR4749 [66] and SPC/E for water [67]. All the molecular systems were centered in a cubic box of edge 9.0 nm^3^. Simulations were performed by following the procedures described in [68,69]. Briefly, boxes were filled with water molecules to have a minimum hydration level of 9 g water/g protein. The ionization states of protein residues were fixed at pH 7, and Cl^−^ or Na^+^ ions were added to keep the system electrically neutral. In particular, 36 Na^+^ were added to all the HSA/miR4749 complexes, while 15 Na^+^ and 21 Na^+^ were added to HSA and to miR4749, respectively. In all the cases, systems with more than 72,000 total atoms were obtained. H bonds were constrained with the LINCS algorithm [70]. The particle mesh Ewald (PME) method [71,72] was applied to calculate the electrostatic interactions with a lattice constant of 0.12 nm. Periodic boundary conditions in the NPT ensemble with T = 298 K and *p* = 1 bar, with a time step of 1 fs, were used. The temperature was controlled by the Nosé–Hoover thermostat with a coupling time constant t_T_ = 0.1 ps [73], while Parrinello–Rahman extended-ensemble, with a time constant t_P_ = 2.0 ps, was used to control pressure [74]. Each system was minimized and then heated to 298 K with steps at 50 K, 100 K 150 K and 250 K. Each system was preliminarily submitted to 10 ns for relaxation, then, it was submitted to 10 ns long MD trajectory, replicated three times, for data collection. Each replica was obtained by randomly changing the initial atom velocities using a Maxwell–Boltzmann distribution at the corresponding absolute temperature, as implemented in the GROMACS package. The temporal evolution of the trajectories was monitored by analyzing the root mean square displacement (RMSD), the root mean square fluctuation (RMSF), and the solvent accessible surface areas (SASA) through the GROMACS package tools [65].

### 3.6. Calculation of the Binding Free Energy

The binding free energy, ΔG_B_, of the HSA/miR4749 complex was evaluated by the Molecular Mechanics Poisson–Boltzmann Surface Area (MM-PBSA) method by following the procedure as reported in [75,76,77]. In summary, ΔG_B_ was estimated from: ΔG_B_ = ΔG_complex_ − (ΔG_receptor_ + ΔG_ligand_), with the free energy, G, given by: G = E_MM_ − TS_MM_ + G_solv_, where E_MM_ is the internal energy, TS_MM_ is the entropic term and the G_solv_ the solvation contribution, further decomposed into electrostatic (G_polar,solv_) and non-polar (G_nonpolar,solv_) parts [78]. The E_MM_ energy was evaluated from E_MM_ = E_elec_ + E_VdW_, where = E_elec_ is the protein–protein electrostatic and E_VdW_ is the Van der Waals interaction energy. The entropic contribution was estimated by the quasi-harmonic approach, as reported in [79]. G_polar,solv_ was evaluated by numerically solving the Poisson–Boltzmann equation with the Adaptive Poisson–Boltzmann Solver (APBS) software [80], setting a 0.512 × 0.510 × 0.506 Å grid-spacing and using the AMBER03 force field parameters, with a probe radius of 1.4 Å for the dielectric boundary. The dielectric constant was set to 2 for the interior and to 78.5 for water [81]. The nonpolar part of the solvation contribution was evaluated by G_nonpolar,solv_ = γ SASA + β, with γ = 2.27 kJ mol^−1^nm^−2^ and β = 3.84 kJ/mol [82]. For each model of the complex, average free energy was evaluated by taking into consideration 10 snapshots, recorded every 1 ps from the last 1 ns of the 10 ns long MD simulation runs, for each of the 3 replicates.

## 4. Conclusions

Evidence of the capability of HSA to interact with miR4749 was provided by AFS and fluorescence spectroscopy (steady state and time-resolved quenching experiments). We highlighted the formation of a stable HSA/miR4749 complex through an allosteric binding mechanism, with an affinity constant K_A_ of about 10^4^ M^−1^. This intermediate affinity value could be compatible with a miRNA carrier role for HSA and, at the same time, susceptible to easy delivery to the appropriate targets. Time-resolved FRET measurements allowed us to reliably evaluate the distance between the lone Trp of HSA and the suitably labelled miR4749. Such a value was used as an input for molecular dynamics-assisted docking and binding free energy calculations to elicit a reliable topological model for the complex. We moreover found that the mechanisms underlying the complex formation are essentially driven by electrostatic interaction between the partners. The identification of the positively charged HSA pocket able to accommodate miR4749 encourages speculation about the possibility that HSA could also be prone to bind other negatively charged miRNAs. Globally, our results contribute to the understanding of fundamental processes at the basis of the miRNA transport in plasma, where HSA could play an important role as a carrier and by facilitating delivery to appropriate targets.

## Figures and Tables

**Figure 1 ijms-23-01291-f001:**
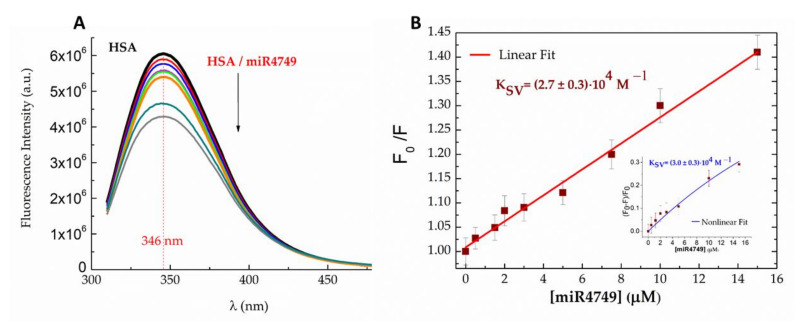
(**A**) Fluorescence emission spectra of 5 µM HSA in PBS buffer (pH 7.4), alone (black line) and with increasing concentrations of miR4749 (0.5–15 µM, coloured lines) at 298K; λ_exc_ = 295 nm. *(***B**) Linear Stern–Volmer plot of the fluorescence quenching of HSA as a function of miR4749 concentration (dark red squares), fitted by Equation (3) (red line); the extracted K_SV_ value being reported. Inset: Nonlinear Stern–Volmer plot of the fluorescence quenching of HSA as a function of miR4749 concentration (dark red squares), fitted by Equation (4) (blue line); the extracted K_SV_ value being reported.

**Figure 2 ijms-23-01291-f002:**
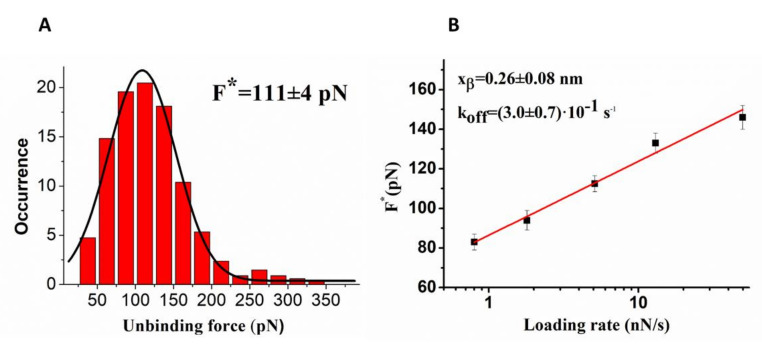
(**A**) Histogram of the unbinding forces for the HSA/miR4749 complex from AFS measurements carried out at a loading rate of 5 nN/s. The most probable unbinding force value (F*) was determined from the maximum of the main peak of the histogram by fitting with a Gaussian function (black curve). (**B**) Plot of the most probable unbinding force, F*, vs. the logarithm of the loading rate for the HSA/miR4749 complex. Red continuous line is the best fit by the Bell–Evans model (Equation (5)); the extracted values for the k_off_ and x_β_ parameters are reported.

**Figure 3 ijms-23-01291-f003:**
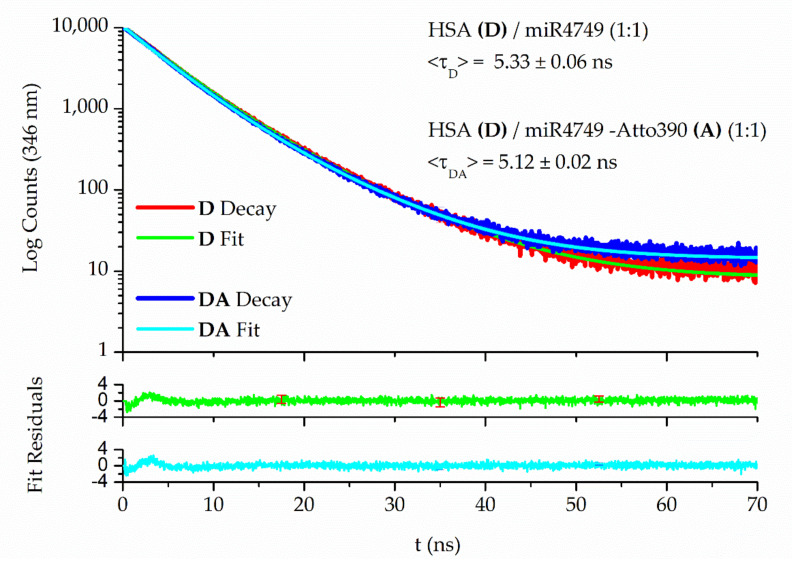
Fluorescence decays of 5 µM HSA/miR4749 (D) and HSA/miR4749–Atto390 (DA) complexes in PBS buffer (pH 7.4), collected through the TC single-photon counting method; λ_exc_ = 295 nm, λ_em_ = 346 nm, at 298 K. The corresponding average fluorescence lifetimes, <τ_D_> and <τ_DA_>, are reported.

**Figure 4 ijms-23-01291-f004:**
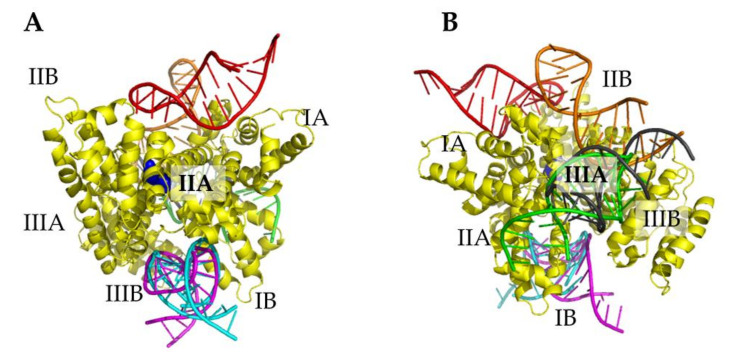
Collective representation of the six best models for HSA/miR4749 complex (**A**) at the front and (**B**) at the back. HSA is coloured in yellow, while miR4749 is coloured as follows: Model 1 (green), Model 2 (orange), Model 3 (magenta), Model 4 (red), Model 5 (gray) and Model 6 (cyan). The regions of HSA are labelled and Trp214 atoms are shown as blue spheres.

**Figure 5 ijms-23-01291-f005:**
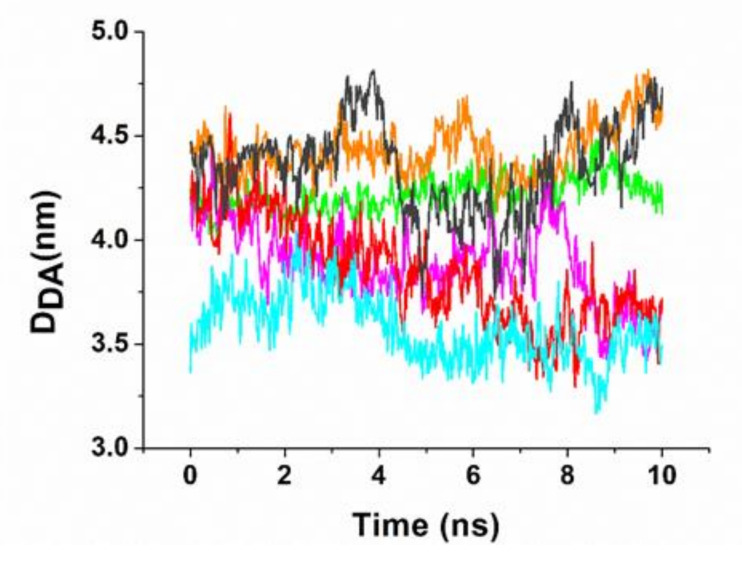
Temporal evolution of the D_DA_ distance between the 5′ end of miR4749 and the aromatic ring center of the lateral chain of Trp214 for the six models of the complex during a MD run. Colours are the same as in Figure 4.

**Figure 6 ijms-23-01291-f006:**
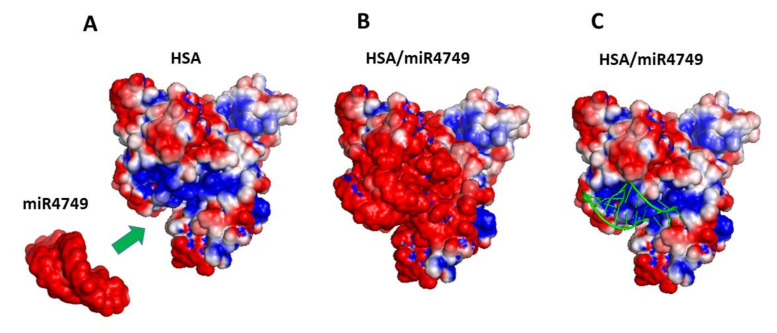
Electrostatic surface potential visualization as evaluated by APBS [54] for the structures of: (**A**) bare HSA and miR4749. (**B**) The HSA/miR4749 complex at the end of a 10-ns long run; (**C**) the HSA/miR4749 complex as in (**B**), in which the skeleton of miR4749 is represented by a green cartoon. Red colour indicates negative charges, while blue colour indicates positive ones.

**Figure 7 ijms-23-01291-f007:**
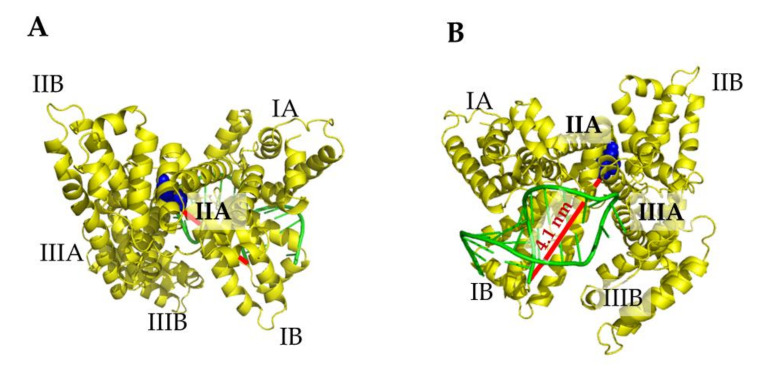
Front (**A**) and back (**B**) view of a graphical representation of the best model (Model 1) for the complex between HSA (yellow) and miR4749 (green). The distance D_DA_ between Trp146 and the 5′ end of miR4749 and the center of the aromatic rings of the lateral chain of Trp214 of HSA (red line) is reported. The regions of HSA are labelled and Trp214 is shown as blue spheres.

**Figure 8 ijms-23-01291-f008:**
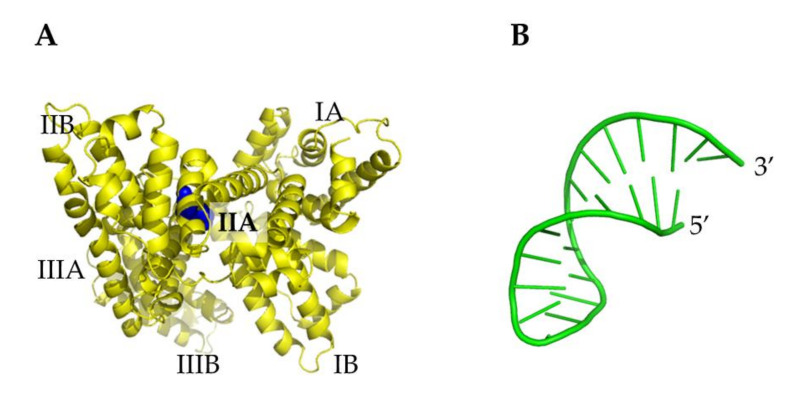
Graphical representation of: (**A**) X-ray structure of HSA monomer (chain A of 1AO6 PDB entry [59]); Trp214 atoms being marked as blue spheres. (**B**) A 3D model of miR4749.

**Table 1 ijms-23-01291-t001:** Properties of the six models for the HSA/miR4749 complex. The distances between the 5′ end of miR4749 and the aromatic ring center of the lateral chain of Trp214 at the beginning (D_DA in_) and the average (D_DA ave_) over the MD simulations are also reported. Binding free energy, ΔG_B_, and its components for the six best HSA/miR4749 interaction models. ΔG_nonpol solv_ represents the nonpolar contribution to the solvation term, ΔE_MM_, the internal energy, −TΔS_MM_, the entropic term and, finally, the ΔG_pol solv_ is the electrostatic contribution to the solvation term.

MODEL #	D_DA in_ (nm)	D_DA ave_ (nm)	ΔG_nonpol solv_ (kJ/mol)	ΔE_MM_ (kJ/mol)	−TΔS (kJ/mol)	ΔG_pol solv_ (kJ/mol)	ΔG_B_ (kJ/mol)
Model 1	4.2	4.1	−50.8	−693	+1223	−2010	−1531
Model 2	4.3	4.4	−33.6	−643	+1202	+1120	+1645
Model 3	3.8	3.9	−64.8	−1298	+1236	+6950	+6823
Model 4	3.8	4.0	−43.8	−439	+1257	−1990	−1216
Model 5	4.3	4.4	−36.8	−378	+1227	+3080	+3892
Model 6	3.7	3.6	−40.1	−1646	+1183	+5690	+5187

## Data Availability

Not applicable.

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
