# Peer review of "Interaction between miR4749 and Human Serum Albumin as Revealed by Fluorescence, FRET, Atomic Force Spectroscopy and Computational Modelling"

_ijms, 2022, doi:10.3390/ijms23031291_

Round 1
Reviewer 1 Report
Authors meritoriously explained the state of the art in miRNA research, its relations with serum albumins, and also the importance to understand basic interactions between miRNAs and unmodified, native serum albumin - HSA used in this research. The miR4749-5p (miR4749) was chosen as a relevant representative of miRNA, based on its biomedical importance.
For the proposed study authors used protein (HSA) intrinsic fluorescence, in particular focusing on the lone Tryptophan (Trp214) of HSA, which served as a solvatochromic fluorophore in fluorescence quenching measurements and as an energy donor in FRET experiments. Obtained results were used in molecular modelling and free energy calculations to propose a binding site or miRNA on the HSA.
In general, the manuscript is nicely written, clearly supporting the scientific importance of the performed study. Applied methods, (both, experimental and computational), are adequately described and performed, thus supporting the Conclusions of the research. After elucidation of two minor issues, the manuscript would be acceptable for publication.
Q1. Authors performed titration of HSA (c=5uM) with miRNA (c=1 – 5 uM) and by fitting results to the Stern–Volmer equation obtained constant KSV of (2.6 ± 0.3) × 104 M-1; which approximately equalled to the affinity constant, Ka. However, the equation for Ka of a simple two-component non-covalent binding event (stoichiometry 1:1), where concentrations of both components are 5uM and Ka=100 uM; dictate that less than 5% of HSA/miRNA complex is formed. Only by increasing the concentration of one component (e.g. RNA) up to 100 uM, more than 50% of HSA/miRNA complex would be formed, and eventually, non-linear emission change could be fitted to the 1st exponential decay, giving directly Ka. Such conditions of a high percentage of HSA/miRNA complex formed would increase the accuracy of other methods applied.
Q2. All experimental results rely on a single spectrophotometric method, based on the one essential fluorophore: lone tryptophan residue of HSA, Trp214. Fluorescence is prone to many artefacts, particularly if one of the emissive species (HSA/miRNA complex) is present in a very low percentage (see Q 1). Thus, it is always recommendable to use another, independent method, at least to estimate the order of magnitude affinity of non-covalent interaction. Isothermal titration calorimetry (ITC) is the ideal method for here studied system since proposed electrostatic and hydrophobic binding interactions should at given concentrations of components (HSA: 5uM, miRNA: 1-100 uM) yield measurable heat change. Such a simple ITC experiment consisting of 3 titrations (1: HSA in a cell, titrated by buffer; 2: buffer in a cell titrated by RNA; 3: HSA in a cell, titrated by RNA) would within one day give a highly useful set of thermodynamic parameters. In the worst case of low heat change of ITC titration, even a single injection ITC experiment would be useful, giving at least total enthalpy change and estimation of Ka.
Reviewer 2 Report
In this study, the interaction of miR4749 was examined with human serum albumin (HSA), employing spectroscopic and modeling studies. It is an interesting work; however, certain details should be discussed regarding the experimentation and data evaluation. Furthermore, some confirmatory measurements are reasonable the support the reliability of the results. Therefore, I suggest the publication of this manuscript only after a major revision. My questions and critical comments are listed below.
Some linguistic corrections are required, and the typos should be corrected. Some examples:
“increase solubilization and circulation” – How can be circulation increased?
“can enhance their half-life” – use “increase” instead of “enhance”
“mir4749”
The ratio of HSA and miR4749 is not higher than 1:1 (both 5 μM). If the Ka value is approximately 10^4 L/mol, then much higher concentrations of miR4749 vs. HSA should be also tested for the more proper determination of the association constant. It also seems to be reasonable because of the large SD values demonstrated in Fig. 2B.
Did Authors correct the inner-filter effect in quenching studies? It has to be performed before the evaluation.
Binding constants have to be determined by non-linear fitting and not only with the SV equation.
It would be reasonable to apply another technique to confirm the binding constant determined in quenching studies (such as fluorescence anisotropy).
Site marker experiments should be performed to support the data from modeling studies.
The resolution of figures (mainly Fig. 2) is poor, it should be improved.
Round 2
Reviewer 2 Report
Authors significantly improved the manuscript. I accept their response and I have no further critical comments.